# A sister lineage of the *Mycobacterium tuberculosis* complex discovered in the African Great Lakes region

Jean Claude Semuto Ngabonziza [1,2,3,16], Chloé Loiseau[4,5,16], Michael Marceau[6,16], Agathe Jouet [7], Fabrizio Menardo [4,5], Oren Tzfadia[2], Rudy Antoine [6], Esdras Belamo Niyigena[1], Wim Mulders [2], Kristina Fissette[2], Maren Diels[8], Cyril Gaudin[7], Stéphanie Duthoy[7], Willy Ssengooba [9], Emmanuel André[10], Michel K. Kaswa[11], Yves Mucyo Habimana [12], Daniela Brites [4,5], Dissou Affolabi[13], Jean Baptiste Mazarati[14], Bouke Catherine de Jong [2], Leen Rigouts [2,3], Sebastien Gagneux [4,5,17✉], Conor Joseph Meehan [2,15,17✉] & Philip Supply [6,17✉]

The human- and animal-adapted lineages of the *Mycobacterium tuberculosis* complex (MTBC) are thought to have expanded from a common progenitor in Africa. However, the molecular events that accompanied this emergence remain largely unknown. Here, we describe two MTBC strains isolated from patients with multidrug resistant tuberculosis, representing an as-yet-unknown lineage, named Lineage 8 (L8), seemingly restricted to the African Great Lakes region. Using genome-based phylogenetic reconstruction, we show that L8 is a sister clade to the known MTBC lineages. Comparison with other complete mycobacterial genomes indicate that the divergence of L8 preceded the loss of the *cobF* genome region - involved in the cobalamin/vitamin B12 synthesis - and gene interruptions in a subsequent common ancestor shared by all other known MTBC lineages. This discovery further supports an East African origin for the MTBC and provides additional molecular clues on the ancestral genome reduction associated with adaptation to a pathogenic lifestyle.

[1] National Reference Laboratory Division, Department of Biomedical Services, Rwanda Biomedical Center, Kigali, Rwanda. [2] Mycobacteriology Unit, Department of Biomedical Sciences, Institute of Tropical Medicine, Antwerp, Belgium. [3] Department of Biomedical Sciences, Antwerp University, Antwerp, Belgium. [4] Swiss Tropical and Public Health Institute, Basel, Switzerland. [5] University of Basel, Basel, Switzerland. [6] Univ. Lille, CNRS, Inserm, CHU Lille, Institut Pasteur de Lille, U1019 - UMR 9017, CIIL - Center for Infection and Immunity of Lille, 59000 Lille, France. [7] Genoscreen, Lille, France. [8] BCCM/ITM, Mycobacterial Culture Collection, Institute of Tropical Medicine, Antwerp, Belgium. [9] Department of Medical Microbiology, College of Health Sciences, Makerere University, Kampala, Uganda. [10] Laboratory of Clinical Bacteriology and Mycology, Department of Microbiology and Immunology, KU LeuvenLeuven, Belgium. [11] National Tuberculosis Program, Kinshasa, DR, Congo. [12] Tuberculosis and Other Respiratory Diseases Division, Institute of HIV/AIDS Disease Prevention and Control, Rwanda Biomedical Center, Kigali, Rwanda. [13] Laboratoire de Référence des Mycobactéries, Cotonou, Benin. [14] Department of Biomedical Services, Rwanda Biomedical Center, Kigali, Rwanda. [15] School of Chemistry and Biosciences, University of Bradford, Bradford, UK. [16] These authors contributed equally: Jean Claude Semuto Ngabonziza, Chloé Loiseau, Michael Marceau. [17] These authors jointly supervised this work: Sebastien Gagneux, Conor Joseph Meehan, Philip Supply. ✉email: sebastien.gagneux@swisstph.ch; c.meehan2@bradford.ac.uk; philip.supply@ibl.cnrs.fr

Tuberculosis (TB), caused by members of the *Mycobacterium tuberculosis* complex (MTBC), is among the ancient scourges of humankind[1], and remains the leading cause of mortality globally due to an infectious disease[2]. Intense research has been dedicated to decipher the evolutionary history of the MTBC and to understand the causes underlying the worldwide spread of TB[3–5]. Current genome data show that the MTBC is comprised of the five human-adapted lineages representing *M. tuberculosis sensu stricto* (L1–4 and L7), two other human-adapted lineages traditionally referred to as *M. africanum* (L5-6) and at least nine animal-adapted lineages[6]. Africa is the only continent where all MTBC lineages are present, suggesting that the MTBC emerged from a common ancestor therein and then expanded to the rest of the world following human migrations[3,7–10]. However, the genomic traits of this common ancestor and the region from which this expansion took place in Africa remain unknown. Whole-genome sequencing (WGS) analyses showed that rare human TB bacilli with a smooth colony morphotype, highly restricted to the Horn of Africa and named *Mycobacterium canettii* (alias smooth tubercle bacilli or STB) represent early evolutionary branching lineages that predate the emergence of the most recent common ancestor (MRCA) of the MTBC (or of the rest of the MTBC, if *M. canettii* is considered to be part of the complex)[4,11,12]. Indeed, whereas known MTBC strains differ by no more than ~2000 single nucleotide polymorphisms (SNPs)[13], *M. canettii* strains are 10- to 25-fold more genetically diverse and separated by at least 14,000 SNPs from the hitherto known MTBC MRCA[4,12]. Moreover, *M. canettii* strains are less virulent and possess highly mosaic genomes, possibly reflecting adaptation to an environmental reservoir favouring active lateral gene flow[4,14–16]. These biological differences support the existence of lineages that reflect intermediate stages in the evolution from *M. canettii* towards the obligate MTBC pathogens.

Here, we describe two exceptional strains representing a new clade, diverging before the MRCA of the other MTBC lineages. These two strains were discovered in two independent analyses and were both multidrug-resistant (MDR; i.e. resistant to at least rifampicin and isoniazid). One was isolated from a TB patient in Rwanda through an ongoing MDR-TB diagnostic trial in Africa. The second isolate came from a patient in Uganda, and was discovered upon screening publicly available draft genome datasets, where it was misclassified as an *M. bovis* strain. We used PacBio and Illumina WGS to reconstruct the full circular genome of the Rwandan strain. We used these data and the available Illumina sequencing data of the Ugandan strain to reconstitute the phylogeny of this novel lineage, which we named Lineage 8 (L8), and further investigate molecular and evolutionary events associated with the emergence of the MTBC.

## Results

**Patient with the L8 MTBC strain in Rwanda.** The Rwandan strain was isolated from a male patient, aged 77 years, HIV-negative, resident of Rulindo district bordering with the Southwest of Uganda, and who had lived in Uganda previously. The patient was diagnosed with rifampicin-resistant TB based on standard Xpert MTB/RIF testing (Xpert; Cepheid, Sunnyvale, CA, USA), which probes for mutations in the rifampicin resistance-determining region of the *rpoB* gene of the MTBC[17]. The results of the assay showed a rare delayed probe B reaction (~3% prevalence in Rwanda)[18], presumed (and later confirmed; see below) to be due to the rifampicin resistance-associated Asp435Tyr mutation[19].

Per routine practice, the patient was initiated on standard short-course MDR-TB treatment (i.e. 9-month WHO-endorsed MDR-TB regimen, including moxifloxacin, kanamycin, protionamide,

ethambutol, clofazimine, high dose isoniazid and pyrazinamide)[20]. However, the patient developed hypotension, and eventually died due to probable cardiac failure, after 20 days of treatment. Phenotypic drug-susceptibility testing (DST) confirmed resistance to both rifampicin and isoniazid, and susceptibility to other anti-TB drugs including ethambutol, fluoroquinolones and second-line injectables.

**Growth characteristics and biochemical properties of the Rwandan L8 strain.** The Rwanda strain was grown in 12.5 days on Mycobacterial Growth Indicator Tubes. Colonies were observed on the fifth week after initial inoculation on Löwenstein–Jensen medium, indicating a slow grower phenotype with rough colonies (Fig. 1). The strain also displayed archetypal biochemical characteristics of *M. tuberculosis sensu stricto*, including niacin production combined with urease hydrolysis (Table 1).

**Genotypic resistance and SNP profile of the Rwandan L8 strain by Deeplex-MycTB.** Following the MDR-TB diagnosis, the strain was included in the first set of tests for an ongoing MDR-TB diagnostic trial 'DIAgnostics for MDR-TB in Africa (DIAMA) Clinicaltrials.gov, NCT03303963', evaluating a new targeted deep sequencing assay, called Deeplex-MycTB. Deeplex-MycTB testing confirmed the presence of the *rpoB* Asp435Tyr mutation conferring rifampicin resistance, along with the *inhA* Ser94Ala mutation conferring isoniazid resistance, consistently with the MDR phenotype identified by phenotypic DST (Fig. 2). This strain also harboured two alleles in phylogenetic positions in *embB* (Ala378) and *gidB* (Ala205) not associated with resistance to ethambutol or streptomycin, which were both shared by several MTBC lineages (L1, 5, 6, 7 and animal lineages) and *M. canettii*[21]. In addition, eight other—so far uncharacterised non-synonymous SNPs were identified in six of the 18 gene targets interrogated by the assay (Fig. 2). Moreover, this test detected an atypical spoligotype pattern, 1111110000000000000000000000000001110000000 (Fig. 2), which was further confirmed by conventional membrane-based spoligotyping testing. This spoligotype pattern was unique in the global spoligotype database that comprises 111,637 MTBC isolates from 131 countries[22].

**WGS analysis and phylogenetic position of the Rwandan and Ugandan L8 strains.** Results from WGS analysis of the Rwandan strain using Illumina sequencing confirmed all Deeplex-MycTB findings.

The strain isolated in Uganda was discovered independently upon screening global MTBC genome datasets publicly available on NCBI/EBI, comprising ~20,000 genomes in total. From subsequent processing with our WGS analysis pipeline, we found 1 genome that did not classify in any of the 7 human-adapted lineages or 9 animal-adapted ecotypes known at the time, but had been misclassified as *M. bovis* isolated from a human patient[23]. These WGS data revealed a similar spoligotype 1111110000000000000000000000000001111000111, with an overlapping signature characterised by the presence of spacers 1 to 5 and 34 to 37 (vs 34–36 in the Rwandan strain) with all intervening spacers missing. Moreover, the Ugandan strain also shared the same *rpoB* Asp435Tyr and *inhA* Ser94Ala mutations and the same sequence alleles in *embB* and *gidB*. The Ugandan strain contained an additional *katG* Ser315Thr mutation conferring high-level isoniazid resistance, as well as the *embA* C-11A and *embB* Asp328Tyr mutations predictive of ethambutol resistance and two *pncA* missense mutations, predictive of pyrazinamide resistance. Moreover, only three of the nine aforementioned uncharacterised SNPs detected by Deeplex-MycTB were shared between both strains.

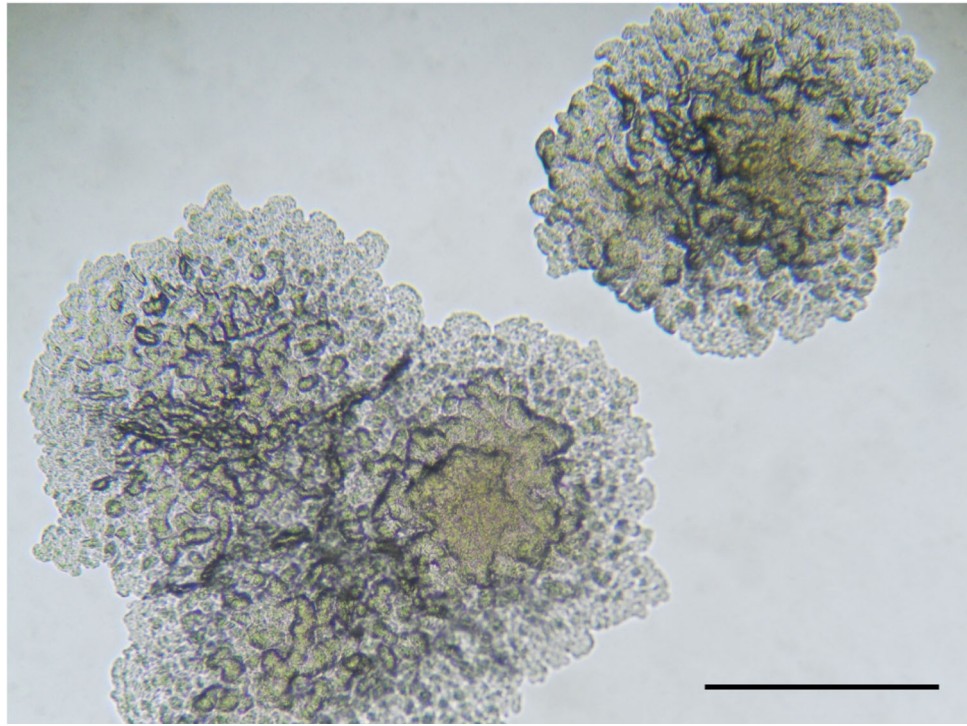

**Fig. 1 Image of colonies from the Rwandan Lineage 8 strain.** Isolate grown on solid medium, with the typical rough morphotype of *Mycobacterium tuberculosis*. Colonies grown on Dubos agar medium were visualized by microscopy. The scale bar represents ~2 mm.

**Table 1 Standard biochemical characteristics of selected mycobacterial species and *M. tuberculosis* complex lineages/ subspecies.**

| Mycobacterial species/ lineage | BCCM/ITM Ref no. | Niacin production | Nitrate reduction | Urease hydrolysis | Tween hydrolysis | Catalase production | Arylsulfatase |
|---|---|---|---|---|---|---|---|
| L8 | 500961 | + | + | + | Weak + | − | − |
| L1 | 500941 | + | + | + | − | Weak + | − |
| L2 | 500945 | + | − | + | − | − | − |
| L3 | 500947 | + | + | + | − | − | − |
| L4 | 500951 | + | + | + | + | − | − |
| L5 | 500953 | + | − | − | + | − | − |
| L6 | 500957 | + | − | Weak + | Weak + | − | − |
| L7 | 500959 | + | − | Weak + | − | − | − |
| *M. bovis* | 500324 | − | − | Weak + | − | − | − |
| *M. bovis* BCG | 500101 | − | − | + | − | Weak+ | − |
| *M. orygis* | 501026 | + | − | + | − | − | − |
| *M. canettii* | 500199 | − | − | + | + | Weak + | − |
| *M. fortuitum* | 500465 | − | + | + | − | Weak + | + |

+ Positive reaction, − negative.

To further assess the relationships between both strains and in comparison to other MTBC strains, a maximum likelihood phylogeny was inferred from 241 MTBC genomes, including representatives of all known human- and animal-adapted lineages[6] and using an *M. canettii* strain as an outgroup. *M. canettii* represents the closest outgroup to the MTBC including L8, as shown by subsequent comparative analysis of a complete L8 genome (see below), and previous observations of ~2.0 Mb larger genomes and substantially lower average nucleotide identities of phylogenetically closest non-tuberculous mycobacterial species such as *M. marinum* and *M. kansasii*[4,24,25]. This reconstruction revealed a unique phylogenetic position of the two new genomes from Rwanda and Uganda (Fig. 3), representing a newly characterised monophyletic clade, in which none of the known

MTBC genomes were contained. A core genome-based phylogeny was also constructed from representatives of the MTBC lineages as well as *M. canettii*, *M. marinum* and *M. kansasii* (Supplementary Fig. 1). This phylogeny confirmed the placement of the L8 Rwandan strain as being a new clade between *M. canettii* and the other lineages of the MTBC. Based on these phylogenies, this clade shares a MRCA with the rest of the MTBC, thus representing a new sister clade to the known MTBC, which we named Lineage 8 (L8). Comprehensive SNP analysis identified a total of 189 SNPs separating both L8 genomes, which is within the range of zero to 700 SNPs found between any two strains within any of the lineages 1 to 7 of the MTBC[13]. On average, two strains of the MTBC including L8 differed by 1443 SNPs (corresponding to 0.04% of the genome, excluding repetitive/'problematic' regions), thus at least an

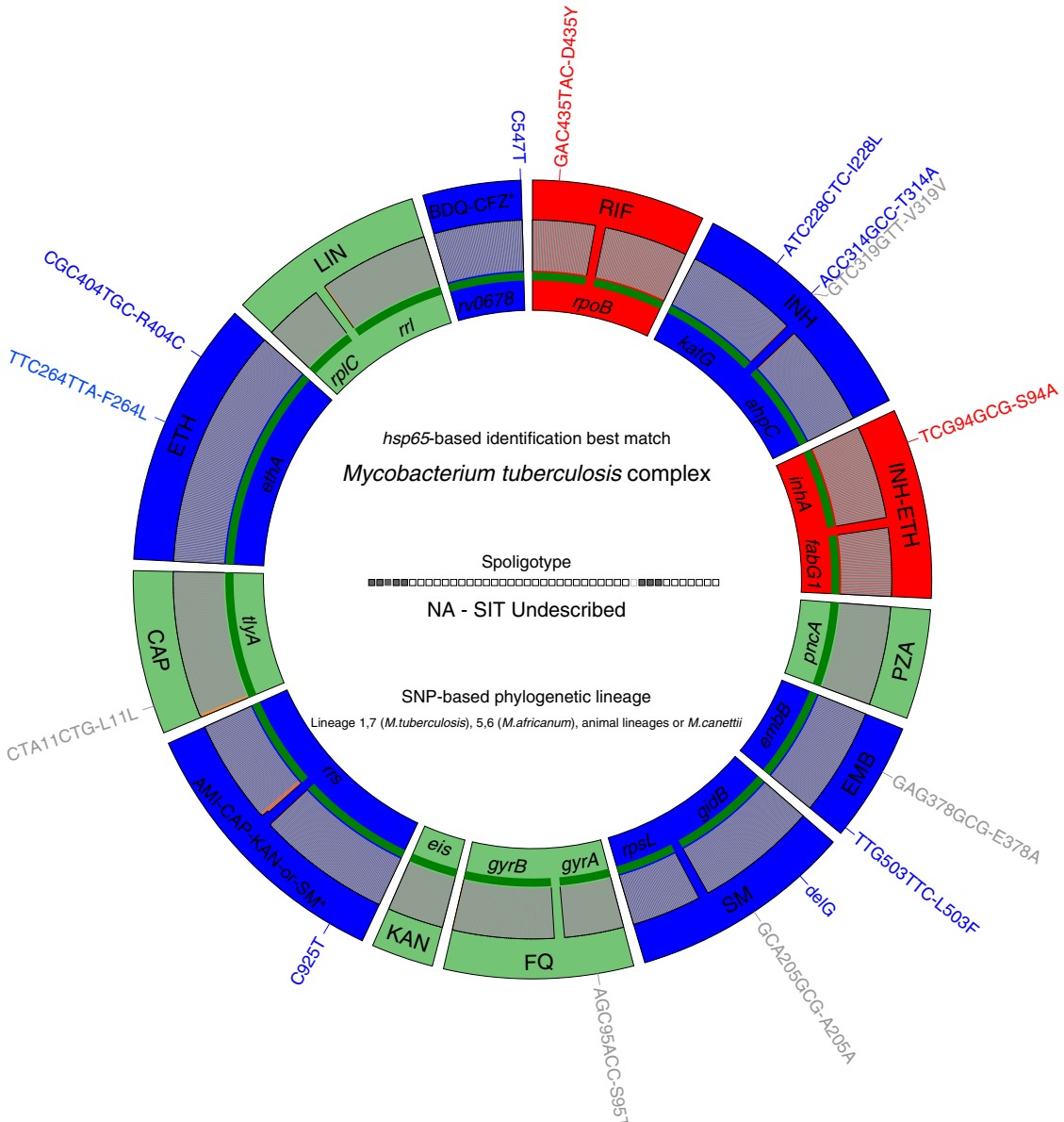

**Fig. 2 Deeplex-MycTB results identifying a MDR-TB strain from Rwanda with an atypical genotypic profile.** Deeplex-MycTB results identifying a MDR-TB strain from Rwanda with an atypical genotypic profile. Target gene regions are grouped within sectors in a circular map according to the tuberculous drug resistance with which they are associated. The two sectors in red indicate regions where rifampicin and isoniazid resistance-associated mutations are detected. The multiple sectors in blue refer to regions, where as yet uncharacterised mutations are detected, whereas sectors in green indicate regions where no mutation or only mutations not associated with resistance (shown in grey around the map) were detected. Green lines above gene names represent the reference sequences with coverage breadth above 95%. Limits of detection (LOD) of potential heteroresistance (reflected by subpopulations of reads bearing a mutation), depending on the coverage depths over target sequence positions, are represented by grey (LOD 3%) and orange zones (variable LOD > 3–80%; only seen in extremities of a few targets, such as the two *rrs* regions) above the reference sequences within the sectors. Information on an unrecognised spoligotype, an equivocal SNP-based phylogenetic lineage and on mycobacterial species identification, based on *hsp65* sequence best match, are shown in the centre of the circle. AMI amikacin, BDQ bedaquiline, CAP capreomycin, CFZ clofazimine, EMB ethambutol, ETH ethionamide, FQ fluoroquinolones, KAN kanamycin, LIN linezolid, INH isoniazid, PZA pyrazinamide, RIF rifampin, SM streptomycin, SIT spoligotype international type.

order of magnitude lower than the SNP distance separating any MTBC strain from *M. canettii*[4,13].

The absence of any matching pattern in the global spoligotype database, as well as the lack of detection of this clade in previous large WGS datasets of MTBC strains from global sources, suggests that L8 is rare and seemingly geographically restricted to the African Great Lakes region. Specifically, the L8 spoligotype signature and the three SNPs shared by both L8 strains were not detected in any of 115 MTBC genomes from a previous drug resistance survey in Uganda[26], nor in 380 rifampicin-resistant strains from Rwanda collected between 1991 and 2018, from routine drug resistance surveillance as well as various drug resistance surveys[27–29]. Furthermore, among 14 other available isolates out of 27 from Uganda and Rwanda tested by Gene Xpert MTB/RIF that showed the same delayed probe B as L8, none displayed the L8 signatures when tested by Deeplex-MycTB or by classical spoligotyping. Likewise, none of >1500 clinical samples from TB patients tested by Deeplex-MycTB from a

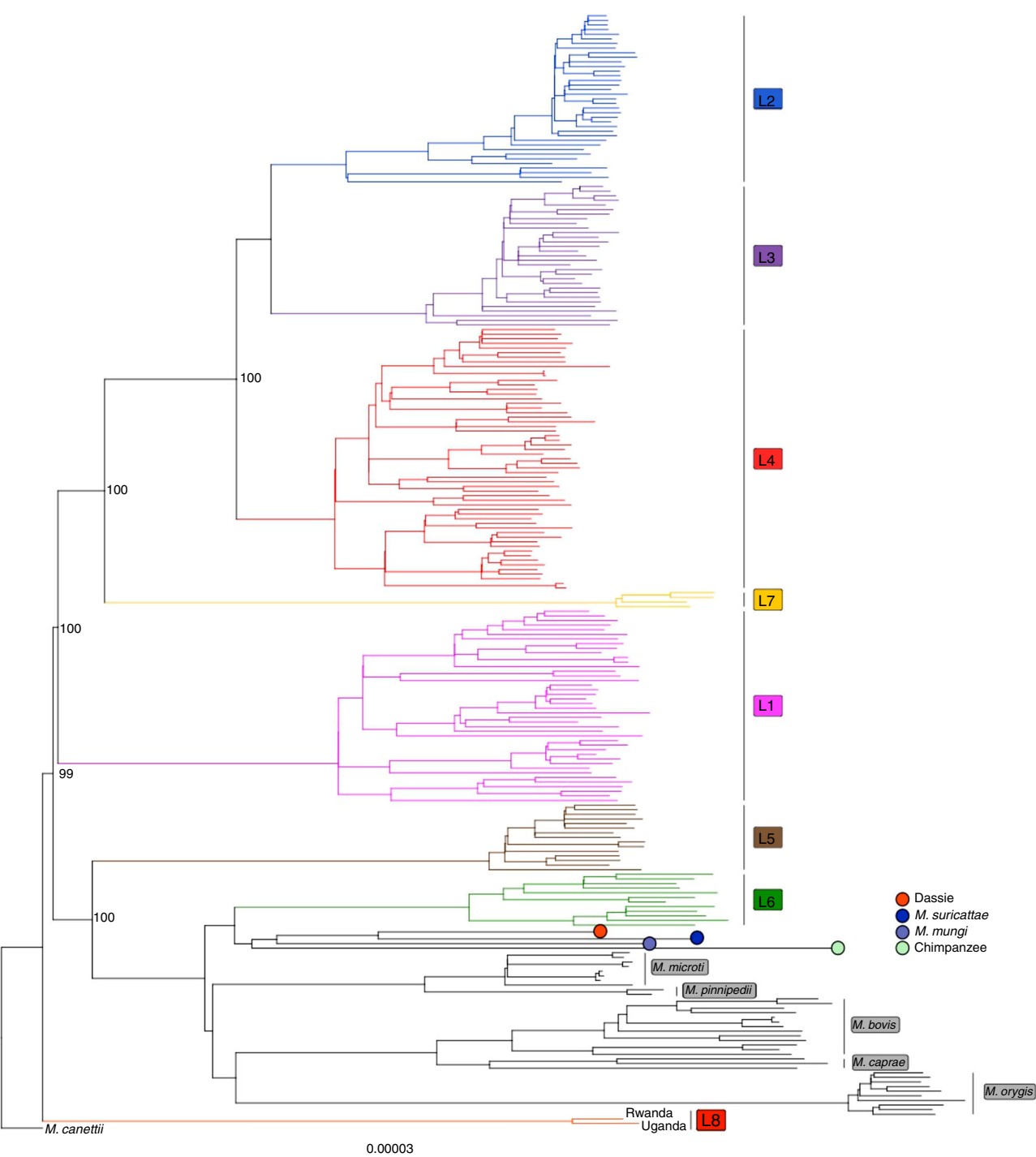

**Fig. 3 Maximum likelihood phylogeny of 241 MTBC genomes, inferred from 43,442 variable positions.** Maximum likelihood phylogeny of 241 MTBC genomes, inferred from 43,442 variable positions. The scale bar indicates the number of substitutions per polymorphic site. Branches corresponding to human-adapted strains are coloured and branches corresponding to animal-adapted strains are depicted in black. The phylogeny is rooted on *M. canettii* and bootstrap values are shown for the most important splits.

recent nationwide drug resistance survey performed in the Democratic Republic of the Congo displayed the L8 spoligotype signature or the specific SNPs.

**Defining features of a complete L8 genome.** To further assess the phylogenetic position of L8 and its split from the rest of the MTBC, the Rwandan strain was subjected to long read-based PacBio sequencing. Comparison of the obtained assembly with 36 available complete genomes of MTBC members, comprising L1-L4 (including H37Rv), *M. africanum* (L6) and *M. bovis* strains, showed a highly syntenic organisation, with no major structural rearrangement between both groups. Although the assembled L8 genome of 4,379,493 bp was within the 4.34–4.43 Mb size range of the other MTBC genomes, it was 30 kb smaller than the 4.41-Mb mean size of genomes of *M. tuberculosis sensu stricto*[30]. However, the largest part of this gap was accounted by the absence of three genomic regions in L8, corresponding to regions

of difference (RDs) known to be variably present or absent in other MTBC (sub)lineages[31,32] (Supplementary Data 1). These include a 9.3 kb PhiRv1 prophage region (RD3), as well as 10.0 kb and 8.5 kb segments corresponding to RD14 and RD5, comprising the *plcD* gene region and the *plcABC* gene cluster, respectively[31]. In L8, each of the two latter regions only contained one copy of the IS*6110* insertion sequence, devoid of direct repeats (DRs) that normally flank IS*6110* copies after transposition, indicating that these deletions in L8 resulted from recombination between two adjacent IS*6110* copies with loss of the intervening sequences[33]. These mobile DNA-related deletions, which also arose independently in several other MTBC branches[31,34], probably occurred after the divergence of L8 from the other MTBC lineages. Apart from these three deletions and two dozen repetitive/multicopy genes (IS6110-related, PE/PPE-, or Mce-encoding), we only found 5 non-repetitive genes, included in two small segments (3.4 kb and 4.4 kb), which were undetected in the complete L8 genome while being present in reference MTBC genomes (Supplementary Data 1).

Conversely, a 4.4 kb genome region was present in the genomes of both L8 strains and in *M. canettii*, but absent in the 36 available complete genomes of MTBC members (Supplementary Data 2). This region comprises the *cobF* gene (Fig. 4), encoding the precorrin 6A synthase involved in the cobalamin/vitamin B12 synthesis, along with two other genes, respectively encoding a PE-PGRS protein

family member and a protein of unknown function. This region shared by the L8 and the *M. canettii* genomes is also present in the phylogenetically proximal non-tuberculous mycobacterial species *M. marinum* and *M. kansasii* (Fig. 4). Via BLAST analysis, we further confirmed the systematic absence of *cobF* in any of 6456 quality draft genome assemblies available as of January 2020 from the NCBI, from strains belonging to lineages 1–7 or the animal lineages of the MTBC. Moreover, we thereby determined that the junction between the sequences flanking the *cobF* deletion was at the same nucleotide position in all but 6 of these genomes, resulting in the truncation of *rv0943c* and *rv0944* genes as seen in the complete MTBC genomes (Fig. 4). Consistent with the clonal evolution of the MTBC with negligible, if any, horizontal gene transfer between strains[1,14,32,35], the perfect conservation of this sequence junction suggests that *cobF* was lost in the MRCA of the other MTBC lineages, after its divergence from L8. The 6 exceptions were 3 strains from lineage 4.3 and 3 strains from lineage 3 that showed slightly larger deletions, including the 5′ region of *rv0943c* or the 5′ region of *rv0943c*, *rv0944* and the 5′ region of *rv0945*, respectively, suggesting probable additional deletion events in particular sub-branches of these lineages.

In contrast, none of the almost 900 other genes specifically identified in the *M. canettii* genomes, and absent in the other MTBC genomes, were found in the L8 genome. The latter finding thus supports the close relationship with the previously known

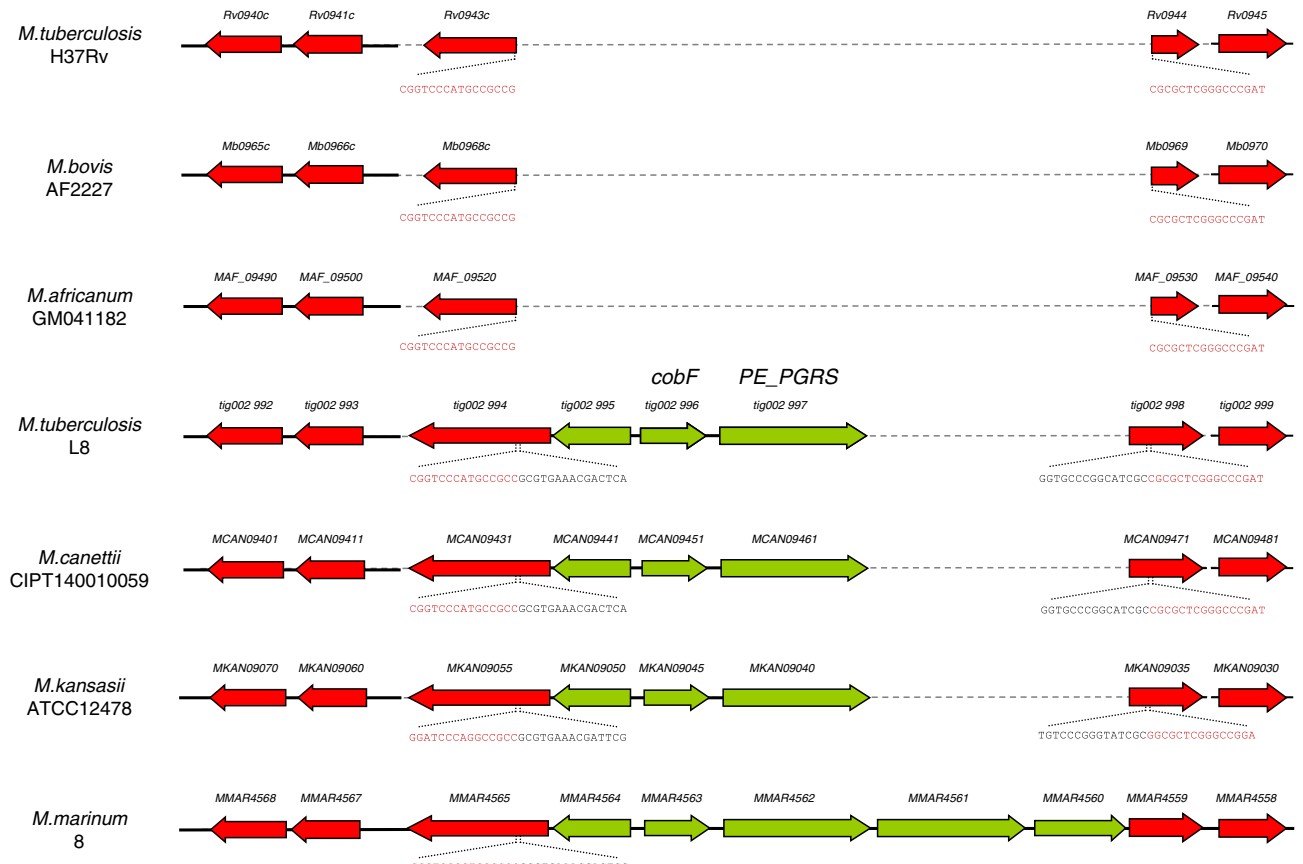

**Fig. 4 Differential presence of the *cobF* region in specific *Mycobacterium* spp. genomes.** Aligned genome segments showing the *cobF* gene region in *M. tuberculosis* L8, *M. canettii* CIPT140010059 (alias STB-A), *M. kansasii* ATCC12478 and *M. marinum* M strains, and the corresponding deletion in *M. tuberculosis* H37Rv, *M. bovis* AF2122/97 and *M. africanum* GM041182. Coding sequences of this region are shown in green, and flanking coding sequences in red. Sequences flanking the deletion point in truncated genes in *M. tuberculosis*, *M. africanum* and *M. bovis*, and in the *cobF* region present in L8, *M. canettii*, *M. kansasii* and *M. marinum* are indicated in red and black, respectively. Dashed lines correspond to missing segment parts relatively to the longest segment found in *M. marinum*.

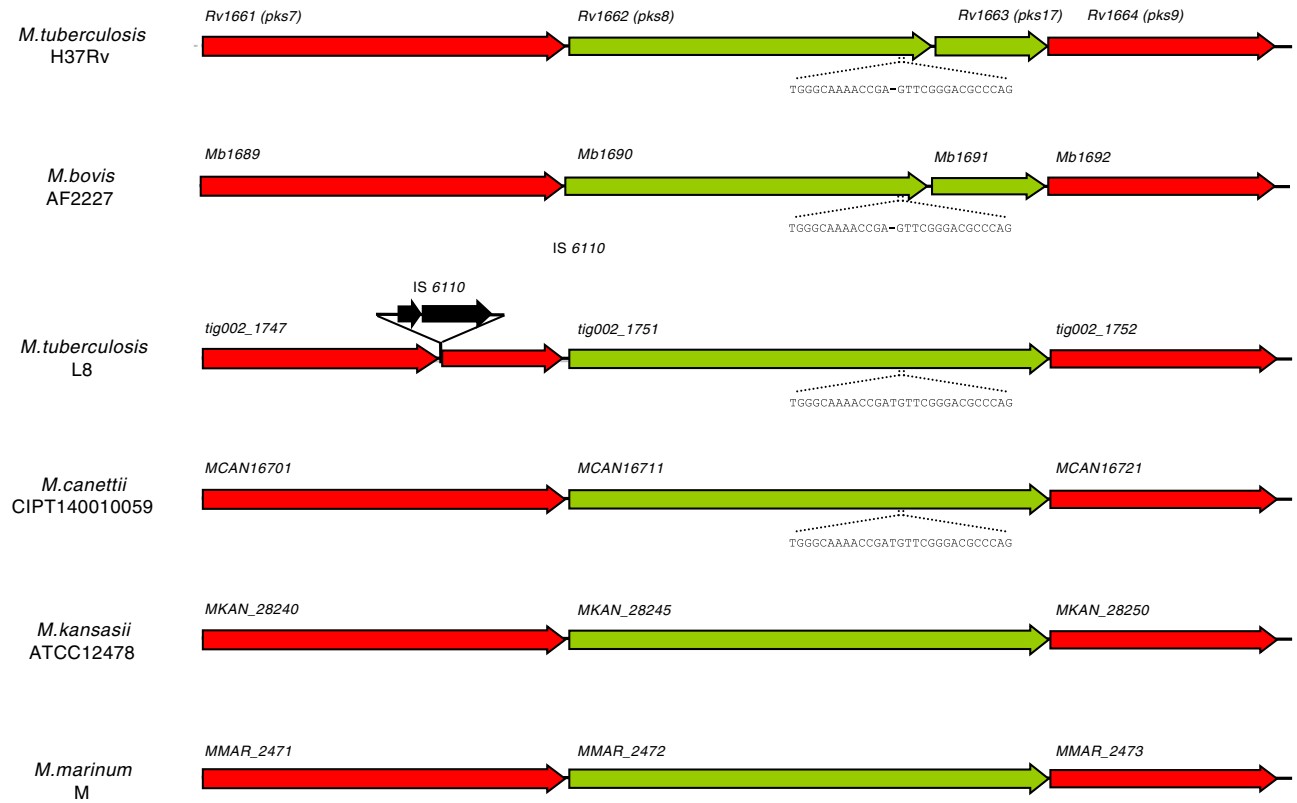

**Fig. 5 Architecture of the *pks8/17* region of specific *Mycobacterium* spp. genomes.** Aligned genome segments showing the interrupted coding sequences *pks8/17* in *M. tuberculosis* H37Rv and *M. bovis* AF2122/97, and complete *pks8* genes in L8, *M. canettii* CIPT140010059 (alias STB-A) and *M. kansasii* ATCC12478. The coding sequences of *pks8/17* and *pks8* are shown in green, and flanking genes in red. Sequences flanking the 1-nucleotide deletion and resulting in a frameshift in *M. tuberculosis* complex strains are indicated.

MTBC lineages indicated by the SNP-based phylogeny, as well as the outgroup position of *M. canettii* relatively to the MTBC including L8.

Further evidence for the early branching of L8 relative to the rest of the MTBC comes from examination of interrupted coding sequences (ICDSs). These ICDSs correspond to frameshifts or in-frame stop codons detected in genes that are intact in a common progenitor, thus putatively representing so-called molecular scars inherited during progressive pseudogenisation of the MTBC genomes[36,37]. Four orthologues of MTBC ICDSs were previously found to be intact in the genomes of *M. canettii* strains, as well as in *M. marinum* and *M. kansasii*[4]. One of these four orthologues (*pks8*), which belongs to a multigene family encoding polyketide synthases involved in the biosynthesis of important cell envelope lipids[38], was also intact in the genomes of both L8 strains (Fig. 5 and Supplementary Data 3). Moreover, we found an additional orthologue of MTBC ICDSs (i.e. *rv3899c-rv3900c*), coding for a conserved hypothetical protein, which was intact in the genomes of *M. canettii*, *M. kansasii*, *M. marinum* and both L8 strains (Supplementary Data 3). These two molecular scars were also likely acquired by the other MTBC lineages after their divergence from the common progenitor shared with L8.

The assembled L8 genome also included 35 of the 50 genes (the exceptions are rv3513c encoding the probable fatty-Acid-Coa ligase FadD18 and 14 genes in the PhiRv1 region; see above) present in MTBC members but not found in any of the *M. canettii* genomes, including a number of genes putatively acquired through horizontal gene transfer by the common ancestor of the MTBC after its separation from *M. canettii*[4] (Supplementary Data 4). This observation additionally supports both the close relationship with the previously known MTBC lineages and the outgroup position of *M. canettii* relatively to the MTBC including L8. Likewise, consistent with the rough colony morphotype of the Rwandan strain, both L8 strains displayed the single polyketide-synthase-encoding *pks5* gene configuration shared by all MTBC members, instead of the dual *pks5* conformation found in *M. canettii* strains involved in the smooth colony phenotype of the latter strains[15]. Thus, the recombination between the two *pks5* genes and the loss of the intervening *pap* gene, thought to have resulted in surface remodelling and incremental gain of virulence after the phylogenetic separation from *M. canettii*[15], already existed in the common progenitor of L8 and the rest of the MTBC. Moreover, both L8 strains also contained the intact TbD1 and RD9 regions, shared by the other 'ancestral' *M. tuberculosis* lineages (L1, L7) but subsequently lost by the so-called 'modern' lineages of *M. tuberculosis* (TbD1 lost in L2-4), *M. africanum* (L5 and L6) and the animal lineages (RD9)[31].

In contrast to the highly clonal population structure of the MTBC, *M. canettii* strains are highly recombinogenic, as apparent from mosaic sequence arrangements in their genomes and functional DNA transfer between *M. canettii* strains mediated by a distributive conjugal transfer (DCT)-like mechanism[4,39]. However, no significant genome-wide recombination signal was detected by ClonalFrameML analysis[40] between L8 and other MTBC strains (Supplementary Fig. 2). In particular, and in contrast to the numerous recombination segments in *M. canettii*[4], the complete L8 genome only contained 26 possible recombination segments, yet the longest of these was 607 bp and the average length was 142 bp.

## Discussion

The discovery of L8 provides unique insights into an ancestor of the MTBC that existed after the *pks5*-recombination-mediated

surface remodelling, which occurred after separation of the MTBC MRCA from the *M. canettii* clade, but preceded the loss of the *cobF* region and gene interruptions in a later common ancestor of the other MTBC lineages. The seeming restriction of this lineage to the African Great Lakes region represents new evidence supporting an origin for the MTBC in the eastern part of the African continent. These findings reinforce results from previous work suggesting an East—rather than a West African origin of the MTBC[3,4,9,11,41].

A distinct ecological niche, linked to a potential environmental reservoir, has been hypothesised to explain the marked geographic restriction of *M. canettii* strains to the Horn of Africa, the lower persistence of these strains in infection models as well as their genome mosaicism implying multiple DNA recombination events within the *M. canettii* strain pool[4,12]. However, although our analysis is limited to two genomes identified to date, our results suggest that L8 is as clonal as the rest of the MTBC[3,14,34,42]. Moreover, the observation that both L8 strains share two uncommon rifampicin- and isoniazid-resistance conferring mutations in *rpoB* and *inhA* suggests that multidrug resistance was already acquired in the common ancestor of these two strains. Isoniazid and rifampicin were introduced in TB treatments in the African Great Lakes region in the late fifties and 1983, respectively (Dr. Armand Van Deun, personal communication). These shared MDR-defining mutations, and the detection of these isolates in human patients in both cases (with reported absence of previous TB history for the Rwandan patient), suggest that these patients were infected with an already-resistant strain, which was exposed to drug selective pressure already decades ago and had been likely circulating in the community for some time. Overall, this pattern thus suggests human-to-human transmission rather than infection from a non-human source. Although based on only two initial strains, these results are consistent with the presumed scenario of a human rather than a zoonotic origin for the MTBC[31,43].

Of note, given the above timeline of introduction of rifampicin and isoniazid in both countries, the ~100 SNPs distance separating these two strains from their MRCA would imply a rapid molecular clock for L8, in the range of the upper bound of 2.2 SNPs/genome/year most recently estimated for other MTBC clades[44]. However, this mutation rate cannot be confirmed until additional L8 samples are uncovered.

Remarkably, the absence of other L8 strains in datasets from Uganda, Rwanda and DRC, together comprising more than 2000 strains, suggests that L8 is rare even within the African Great Lakes region. Such scarcity is compatible with selective sweeps of later branching MTBC strains, introduced more recently into the region. Similar scenarios have also been proposed to explain the slow apparent replacement of MTBC L5 and L6 by L4 in West Africa[45–47] and the restriction of L7 to Northern Ethiopia[48].

Loss-of-function linked to the deletion of *cobF* is a plausible candidate molecular event involved in such a replacement scenario for L8. Indeed, loss-of-function appears to be an important mechanism driving the pathoadaptive evolution of the TB pathogen, as shown for the role of the loss of lipo-oligosaccharide production (via recombination in the *pks5* locus)[15] in the evolution towards increased virulence from *M. canettii* to MTBC strains. Likewise, loss of secretion of PPE-MPTR and PE_PGRS proteins by the type VII secretion system ESX-5 (via mutations of the *ppe38* locus) has been involved in the hypervirulence of recent branches of L2 (alias 'modern' Beijing) strains[49]. The loss of the *cobF* region in the other MTBC lineages, inferred from comparative genomics with *M. canettii* and non-tuberculous mycobacteria[4], was previously hypothesised to reflect enhanced adaptation to an intracellular parasitic lifestyle[50]. Indeed, the

cobalamin/vitamin B12 synthesis pathway, of which the *cobF*-encoded precorring-6a synthase is a component, represents a highly complex and energy consuming process with about 30 enzymatic steps[51]. Although the absence of this component may not entirely ablate cobalamin biosynthesis[52,53], its loss might have resulted in gain of fitness and reflect enhanced pathogenic professionalisation, by economical reliance upon the mammalian host environment as source of vitamin B12. As an additional plausible but not necessarily mutually exclusive hypothesis, such selective sweep of L8 might have been (further) enhanced by the loss of theTbD1 region in later branching MTBC strains. This region, which we also found intact in L8, as is the case in the 'ancestral' *M. tuberculosis* lineages L1 and L7, encodes members of the mycobacterial membrane protein families MmpL. Very recent findings indicate that the loss of this region in later branching MTBC strains was also associated with a gain of virulence, and the deletion of TbD1 at the origin of the 'modern' *M. tuberculosis* lineages L2/L3/L4 has therefore been suggested as a key driver for their global epidemic spread[54]. If true, more recently emerged or introduced *cobF*- and TbD1-deleted strains might conceivably have largely outcompeted L8 strains. This hypothesis could be explored by assessing the growth and the virulence/fitness in cellular and animal models, of recombinant *cobF*- and/or TbD1-knock-out, as well as *cobF*- and/or TbD1-knock-in strains, derived from the available Rwandan L8 strain and other MTBC strains, respectively.

In conclusion, our genomic data, on an as-yet-unknown ancestral stage between the MTBC and the putative progenitor pool of *M. canettii*-like mycobacteria, thus suggest further experiments to examine candidate molecular events potentially involved in the pathoadaptive evolution of *M. tuberculosis*. The discovery of such rare strains raises the possibility for the existence of further extant strains, especially in Eastern Africa, representing other clades further closing the biological gap between the MTBC and *M. canettii*.

## Methods

**Rwandan patient recruitment and ethics statement**. The patient in Rwanda was recruited into, and gave informed consent for, the DIAMA study, which was approved by the Rwanda National Ethical committee (IRB 00001497 of IORG0001100; Ref No.0069/RNEC/2017). Consent to publish identifying information was obtained from the patient's representative.

**Phenotypic characterisation**. We studied conventional mycobacterial growth and biochemical characteristics including colony morphology, niacin production, nitrate reduction, *p*-nitro benzoic acid growth inhibition, catalase production, urea hydrolysis, Tween 80 hydrolysis and thiophene carboxylic acid hydrazide growth inhibition[55]. For comparative purpose, a reference set of the seven known human-adapted MTBC lineages[56], together with *M. canettii* (BCCM/ITM500199), *M. bovis* (BCCM/ITM500324), *M. bovis* BCG (BCCM/ITM500101) and *M. orygis* (BCCM/ITM501026) strains were processed with the novel strain isolated in Rwanda. Moreover, phenotypic drug-susceptibility testing to first- and second-line anti-TB drugs was done using the proportion method[57]. The strain isolated in Uganda was not available for phenotypic characterisation.

**Targeted- and whole-genome sequencing of the Rwandan strain**. Targeted sequencing was performed by using the Deeplex-MycTB assay[58] (Genoscreen, France). Briefly, this assay relies on a 24-plexed PCR amplification of mycobacterial species identification (*hsp65*), genotyping (spoligotyping and phylogenetic single nucleotide polymorphisms (SNPs)) and 18 *M. tuberculosis* complex drug resistance-associated gene targets. This test and short-read Illumina-based WGS were performed on the Rwandan strain as follows. A bead beating method was used to extract DNA from colonies as follows: a loopful colonies from LJ were suspended in Tris-EDTA buffer, and heat inactivated at 95 °C for 20 min. After cooling at room temperature, the suspension was centrifuged at 20,000×*g* for 30 min followed by discarding supernatant and 250 µl of 10 mM Tris-HCl pH 7.8 were added and briefly vortexed. Mixture was incubated at 95 °C for 15 min, then spun down briefly followed by transferring entire volume in a new microcentrifuge tube containing 0.5 g of zirconium beads (Sigma-Aldrich, St. Louis, USA). For destruction of the solid mycobacterial cell wall, the mixture was vortexed at high speed for at least 30 s followed by briefly spinning down and incubation at −20 °C for at least 30 min. After

thawing at room temperature, the mixture was briefly spun down and 200 μl of supernatant was transferred to a new microcentrifuge tube. For gDNA concentration, 1 μl of glycogen solution (Sigma-Aldrich, St. Louis, USA) was added followed by 0.1 volume of 3 M sodium acetate at pH 5.2 (Thermo Fisher Scientific, Waltham, MA USA), then 3 volume of 100% pre- cooled ethanol were added and vigorously vortexed for 10 s. The mixture was incubated at −20 °C for 10 min. After thawing at room temperature, the mixture was centrifuged at 15,000 × g for 20 min followed by discarding supernatant, then 600 μl of freshly prepared pre-cooled 70% ethanol was added followed by centrifugation at 15,000 × g for 5 min. Supernatant was discarded and the tube was air dried. gDNA was resuspended in 20 μl of sterile molecular grade water. The yield was measured by Qubit dsDNA BR Assay Kit (Life Technologies, Carlsbad, USA).

Libraries of Deeplex-MycTB amplicons or genome fragments were constructed using the Nextera XT kit and sequenced on an Illumina MiSeq platform with paired end, 150-bp read lengths (Illumina, CA, USA). DNA extraction suitable for PacBio SMRT sequencing was performed using the Genomic DNA Buffer Set (Qiagen Inc, Germantown, Maryland, USA) as follows: colonies (70.3 ± 1.0 mg) from a one month old LJ were transferred into a 50 ml falcon tube containing 3.5 ml of the Qiagen buffer B1 and 70 μl of 10 mg/ml RNAse (Life Technologies, Carlsbad, USA) solution followed by thoroughly vortex and heat inactivation of bacilli at 80 °C for 1 h. Vortexing at later stages was avoided to maintain high molecular weight DNA, by gently inverting or swirling instead. After cooling at room temperature, 100 μl of 100 mg/ml lysozyme (Sigma-Aldrich, St. Louis, USA) was added and the tube was incubated at 37 °C for 60 min, followed by adding 1.2 ml of 2.5 mg/ml of Proteinase K (MP Biomedicals, Santa Ana, USA) and again incubated at 37 °C for 60 min. For protein denaturation (nucleases and DNA- binding proteins), 1.2 ml of buffer B2 was added to the mixture and placed overnight in horizontal shaker at 50 °C (30 RPM) alongside with Qiagen buffer QF. On the next day, a Qiagen Genomic-tip 100/G tip was placed over a 50 ml falcon tube and equilibrated with 4 ml of buffer QBT allowing the column to empty by gravity flow. The overnight incubated mixture was centrifuged at 3700 × g, 4 °C for 10 min, followed by applying the supernatant to the equilibrated column allowing it to enter the resin by gravity flow. The column was washed by adding twice 7.5 ml of buffer QC and then placed over a new clean 50 ml falcon tube, and the DNA was eluted with 5 ml pre-warmed QF buffer. For DNA precipitation, 3.5 ml of isopropanol were added followed by centrifugation at 4500 × g, 4 °C for 20 min, then the supernatant was gently discarded. gDNA was then washed with 1.5 ml cold (4 °C) 70% ethanol, centrifuged at 10,000 × g, 4 °C for 15 min followed by discarding all supernatant and drying the pellet in the Speedvac for 5 min at the medium drying setting. Finally, 200 μl of 10 mM Tris-Cl, pH 8.0 buffer was added and mixed gently by tapping and refrigeration overnight. On the next day, gDNA pellet was resuspended in the horizontal shaker at 65 °C (250 RPM) for 15 min. The purity and integrity of the gDNA were examined through 0.5% agarose gel electrophoresis and yield was measured using the NanoDrop 1000 Spectrophotometer (Thermo Fisher Scientific, Waltham, MA USA) and Qubit dsDNA BR Assay Kit (Life Technologies, Carlsbad, USA) according to manufacturer's instructions. Sequencing was performed on a PacBio RS II using the SMRT technology.

**Deeplex-MycTB analysis and spoligotyping**. Analysis of the Deeplex-MycTB sequencing data, including SNP calling and spoligotype identification, was performed by read mapping on *M. tuberculosis* H37Rv sequence references, using a parameterised web application (GenoScreen)[58]. Membrane-based spoligotyping was performed as described previously[59].

**Illumina whole-genome sequencing analysis**. Raw genomic reads from the newly sequenced L8 genome from Rwanda and the L8 genome from Uganda (SAMN02567762) were processed as previously described[60]. Briefly, the reads were trimmed with Trimmomatic v0.33.22[61] and reads larger than 20 bp were kept. The software SeqPrep (https://github.com/jstjohn/SeqPrep) was used to identify and merge any overlapping paired-end reads. The resulting reads were aligned to the reconstructed ancestral sequence of the MTBC[62] using the mem algorithm of BWA v0.7.13[63]. Duplicated reads were marked using the MarkDuplicates module of Picard v2.9.1 (https://github.com/broadinstitute/picard) and local realignment of reads around InDels was performed using the RealignerTargetCreator and Indel-Realigner modules of GATK v3.4.0[64]. SNPs were called with Samtools v1.2 mpileup[65] and VarScan v2.4.1[66] using the following thresholds: minimum mapping quality of 20, minimum base quality at a position of 20, minimum read depth at a position of 7X, maximum strand bias for a position 90%.

The spoligotype pattern of the strain from Uganda was extracted in silico from the raw reads using KvarQ[67].

**Phylogenetic reconstruction**. The maximum likelihood phylogeny was inferred with RAxML v.8.2.8[68] using an alignment containing only polymorphic sites and the branch lengths of the tree were rescaled using invariant sites[44,69].

A position was considered polymorphic if at least one genome had a SNP at that position. Deletions and positions not called according to the minimum threshold of 7× were encoded as gaps. We excluded positions with more than 20% missing data, positions falling in PE-PGRS genes, phages, insertion sequences and in regions with

at least 50 bp identity to other regions in the genome. We also excluded variable positions falling in drug resistance-related genes. The phylogeny was computed using the general time-reversible model of sequence evolution (-m GTRCAT -V options), 100 bootstrap inferences and *M. canettii* (SRR011186) were used as an outgroup to root the phylogeny.

**Whole-genome de novo assembly, annotation and comparative genomics**. Raw PacBio reads obtained from the Rwandan strain were assembled with Canu v1.6[70], using default settings and an expected genome size of 4.4 Mbp, typical of MTBC strains. After discarding 60,272 reads below minimal quality parameters, 106,681 reads were used for the assembly. On the basis of the expected genome size, the average coverage depth was estimated at 186× using raw reads, and 39× and 38× using corrected and trimmed reads, respectively. The obtained unique contig of 4,387,285 bp was circularised with Circlator v1.5.5[71] using default settings, resulting in an assembly of 4,379,493 bp. Additional sequence verification and correction was then performed by mapping Illumina reads obtained from the same strain, using pacbio-utils v0.2[72] (https://github.com/douglasgscofield/PacBio-utilities) and snippy v4.4[73] (https://github.com/tseemann/snippy). Alignments of the final assembly were performed against an ensemble of complete genome sequences available from 38 strains of tubercle bacilli. This set included 34 *M. tuberculosis* strains from lineages 1, 2, 3 and 4 (comprising H37Rv), *M. africanum* L6 GM041182, *M. bovis* AF2122/97, as well as the closest STB-A (CIPT140010059) and most distant (STB-K) *M. canettii* strains (Supplementary Data 5). Comparative alignments and genome annotation were performed based on BLAST searches and analysis of gene synteny, using Artemis and Artemis comparison tool[74], as well as a custom Multiple Annotation of Genomes and Differential Analysis (MAGDA) software previously used for annotation of *M. canettii* and *Helicobacter pylori* genomes[4,75]. Comparisons with orthologues from *M. canettii* STB-D, -E, -G, -H, -I and -J in addition to STB-A and -K, and from *M. marinum* type strain M and *M. kansasii* genomes were additionally done using the Microscope platform v3.13.3[76]. When applicable, annotations were transferred from those of *M. tuberculosis* or *M. canettii* orthologs in the TubercuList/Mycobrowser database, using BLAST matches of >90% protein sequence identity, an alignable region of >80% of the shortest protein length in pairwise alignments and visual inspection of the gene synteny. Genome completeness was assessed using CheckM[77] using the lineage-specific workflow and *Mycobacterium* as the genus. The PacBio assembled Rwandan strain was found to have 98.74% completeness and 0% contamination or strain heterogeneity, making it suitable for further analyses.

ACT comparison files were generated using MAUVE 2015-02-25 software to visualise the genome-wide distribution of SNP densities between the assembled L8 genome from Rwanda and *M. tuberculosis* H37Rv and *M. canettii* STB-A and STB-K genomes. Recombination between L8 and other MTBC lineages or *M. canettii* was assessed from a progressive MAUVE alignment of the PacBio assembled L8 genome and previously published closed genomes[73] using ClonalFrameML[40]. To further assess the phylogenetic placement of the Lineage 8 strain, a core gene alignment was constructed using the completed genomes of the Rwandan L8 strain, representatives of the MTBC lineages 1–4 and 6, *M. canettii, M. bovis, M. marinum* (accession GCF_000419315.1) and *M. kansasii* (accession GCF_000157895.3). The GFF files of each genome were input to roary[78] with an 80% identity cut-off, as has been done for previous genus-level mycobacterial core alignments[79]. A phylogenetic tree was constructed from the core gene superalignment using RAxML-NG v0.9.0[80] under the GTR + Gamma model of evolution with 20 starting trees. Bootstrapping was run until autoMRE converged with a value of 0.03 (50 replicates).

**Reporting summary**. Further information on research design is available in the Nature Research Reporting Summary linked to this article.

## Data availability

The complete genome sequence of the L8 strain from Rwanda was deposited in the NCBI repository under project PRJNA598991 with SRR10828835 and SRR10828834 accession codes for Illumina- and PacBio-derived genome sequences, respectively. The strain can be requested from the BCCM/ITM [http://bccm.belspo.be/] collection with accession code ITM-500961.

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

## Acknowledgements

Calculations were partially performed at sciCORE (http://scicore.unibas.ch/) scientific computing core facility at University of Basel. This work was supported by EDCTP2 grant DRIA2014-326—DIAMA of the European Union, the Belgian General Directorate for Development Cooperation (PhD fellowship to J.C.S.N.), Grant ANR-16-CE35-0009 from Agence Nationale de la Recherche, the Swiss National Science Foundation (Grants 310030_188888, IZRJZ3_164171, IZLSZ3_170834 and CRSII5_177163), and the European Research Council (309540-EVODRTB). The views and opinions of authors expressed herein do not necessarily state or reflect those of EDCTP. The funders had no role in study design, data collection and analysis, decision to publish or preparation of the manuscript.

## Author contributions

S.G., P.S., C.M., B.C.d.J., L.R. and J.C.S.N. designed the study. P.S., J.C.S.N., C.L., M.M., C.M. and S.G. analysed data and wrote the manuscript, with comments from all authors. A.J. and C.M. performed the assembly of sequences. M.M. annotated the L8 genome. C.L., F.M., D.B. and A.J. performed SNP analyses and phylogenetic reconstruction. M.M., R.A. and P.S. conducted comparative analyses of complete mycobacterial genomes, with support from C.L. and O.T. J.C.S.N., E.B.N., Y.M.H., J.B.M., W.M., K.F. and M.D. performed and/or analysed data from mycobacterial isolation, growth assays, phenotypic characterisation and/or molecular tests. S.D., C.G., J.C.S.N., E.B.N., E.A., W.S. and M.K.K. conducted targeted deep sequencing analyses and/or provided Xpert data. B.C.d.J and D.A. run the DIAMA trial.

## Competing interests

P.S. was a consultant of Genoscreen; C.G. and S.D. were employees of the same company. The other authors declare no competing interests.
