## [Peer Review File · Nature Communications]

Reviewers' comments:

Reviewer #1 (Remarks to the Author):

Jean Claude Semuto Ngabonziza and co-workers report the discovery of a novel, rare, geographically unique, human-to-human transmitted lineage (L-8) that is a phylogenetic outlier of canonical MTBC complex consisting of two major lineages. Complete genome-based investigation of a strain from this lineage explains the emergence, diversification, and success of MTB as a virulent and successful pathogen. The genomic resource and findings from this lineage fill a major and probably most critical gap in systematic understanding on the East African origin of MTBC from environmental or -non-professional pathogenic form as STB to an obligate, successful and virulent pathogen.

Below are major comments that need to be addressed before taking any decision on the manuscript

- 1) Why only *M. canettii* is used as an out-group? There is no statistical or phylogenetic support for the placement of L-8. Why can't it be part of second branch of MTBC (hence zoonotic origin) or in fact the common origin of *M. canettii* and L-8? The authors need to include additional species from STB (*M. kansasii*, *M. marinum*, etc) as outliers to support the intermediate position of L-8 between *M. canettii* and canonical MTBC.
- 2) How their study explains the origin of human and animal associated lineages...about host jump or about how human transmitted the disease to animals.
- 3) The authors report that "Among 14 other isolates out of 27 from Uganda and Rwanda tested by Gene Xpert MTB/RIF that showed the same delayed probe B as L8, none displayed the L8 signatures when tested by Deeplex Myc-TB or by classical spoligotyping". Since the initial phenotype was same as L-8 lineage strain and considering outlier status of L-8, why authors have not investigated genome of these 14 or in fact 27 isolates or at least tested the presence or absence of *cobF* gene in these 27 isolates?
- 4) What is the prevalence of *cobF* in sub-sample 1500 isolates originating from Uganda, Rwanda and DRC that were screened using Deepflex MTB or in the NCBI WGS data set of strains from MTBC complex?
- 5) In the context of their major finding, what is the status of *cobF* in the draft and the complete genomes of MTBC available in NCBI.
- 6) More importantly, what is the status of *pks8* and *cobF* gene region in Ugandan strain? What is the colony morphology of Ugandan strain?
- 7) Genome quality statistics like completeness and coverage as estimated by softwares like CheckM is lacking.
- 8) In the results section describing complete genome assembly of PacBio reads, coverage of the genome is not clearly mentioned. What does 186x, 39x and 38x infer? Are these corresponding to chromosome and plasmids in complete genome?

Other comments

- 1) In methods the last section it is mentioned as L9 genomes, it should be L8
- 2) Figure 1 legend L9 is mentioned.....it should be L8.

Reviewer #2 (Remarks to the Author):

This is a well-written paper reporting a very interesting and relevant finding for the TB community. In the context of two separate analyses (1 done in an MDR-TB trial, the other a screening of publicly available genomes), authors identified what appear to be a novel lineage of MTBC. The first strain, from Rwanda, was available to the authors for sequencing with PacBio/Illumina to complete the genome. The latter strain was from Uganda; the raw sample did not appear to be available for further analysis. The Rwandan and Ugandan strains appear very similar based on a variety of analyses (e.g., spoligo, comparison of AMR genes/mutations), and form a new clade on phylogenetic analysis compared to a representative set of global TB human and animal-adapted lineages.

Authors presented a comprehensive, detailed, and convincing analysis of these two strains, therefore, most of my comments are fairly minor.

Last paragraph of the Intro – “we used PacBio and Illumina WGS to reconstruct the full circular genome” – please specify which sample (country) was used for this as well as why this was not done for both samples here. I understand only sequencing data was available for the second patient’s sample, but this was only clear much later in the paper; it would be more clear if this were stated upfront.

“Molecular scars”? Could authors please include a definition of this in the current manuscript, as even working in this field, this is not terminology I am familiar with; this definitely needs a bit more explanation to be accessible to a broader audience. Along these lines, will the general audience of this journal know what standard short-course MDR TB treatment is? Please review the manuscript for very TB-specific concepts, and whether these may warrant a bit of additional explanation for those who aren’t TB specialists.

There were a couple places where authors should be less firm about their conclusions, given the paucity of data (2 samples so far from this lineage); for example, in the Abstract, authors state Lineage 8 is restricted to the African Great Lakes region. They also state this on the second page after “WGS analysis” (“indicate that L8 is generally rare and geographically restricted to...”). While their two isolates are from this region, and the work is suggestive that this lineage is from this area, this is not conclusive from this work, so I would change the wording a bit here to convey this uncertainty. Another example is “our results indicate L8 is as clonal as the rest of the MTBC”; again, this is an analysis of 2 samples so this is very preliminary and suggestive at best.

Related to this, please show the ClonalFrameML results in the Supplement, rather than saying ‘data not shown’. Thank you.

Results – “L8 related TB patient in Rwanda” – is it ‘L8 related’ or is it just a ‘patient with L8 TB’? The wording could be changed here.

Figure 2 - it says limits of detection are indicated in grey and orange at each position; this really wasn’t too obvious when looking at the figure – can authors please specify they mean the middle circle or otherwise indicate this in some way on the figure directly?

There is three instances of "L9" being used instead of L8; please correct.

In the Discussion, authors report no previous TB treatment for the Rwandan patient. Have authors tried to contact the publishers of the Ugandan genome to see if this information would be available for this patient?

Also in the Discussion, "If true, more recently emerged or introduced cobF-deleted strains might have conceivably largely outcompeted L8 strains" – how would authors propose to test this hypothesis?

A few comments for clarity:

While I appreciate the format of the journal has the Methods last, many readers prefer to read the Methods before the Results to be able to properly interpret them. As the paper stands, some of the Methods are not clear on their own; for example, authors mention the DeeplexMycTb assay in the Methods, but not what this is and what samples it is being done on, and don't explain that the Rwandan sample is actually avai. Could this please be added to the Methods themselves?

In the Discussion "suggests prolonged exposure to antibiotic treatment and human-to-human transmission of a drug resistant strain" – This sentence could be rephrased for clarity. I think authors mean that these patients were likely infected with already-resistant strains, which had been likely circulating for some time in the community, not that these patients themselves had prolonged exposure to antibiotics? The following two sentences (starting with "The observation that..." could possibly be moved before this.

Please add line numbers and page to submissions if possible. It's more difficult to keep track as a reviewer and properly refer the authors to the right sections without these. An extra space or indenting for new paragraphs is also helpful. Thank you.

Reviewer #3 (Remarks to the Author):

The paper "A sister lineage of the Mycobacterium tuberculosis complex discovered in the African Great Lakes region" constitute a description of a new mycobacterium tuberculosis complex (MTBC) lineage which forms a taxonomic outgroup to the typically pathogenic lineages observed to date, but is more closely related to these and the possibly environmental *M. canettii*. The analyses performed strongly suggest that the new lineage (L8) is a very rare pathogen of humans. The observation is interesting, but in my opinion, of limited interest to people outside the core TB community, and would perhaps be more suitable for a more specialized journal.

The presence of resistance mutations yielding INH and RIF mutations which are both rare, yet common to both L8 isolates is quite interesting, especially in light of the significant SNP-distance between the two isolates (100 SNPs). Yet, it is also clear that the authors don't have much evidence at the moment to explain this observation which they state would indicate a high substitution rate (for the resistance SNPs to have emerged following the introduction of the relevant drugs in the region). In the discussion, the authors don't cite any source for their statement about the history of INH and RIF use in the region – could RIF possibly have been used earlier? Could the isolates have emerged elsewhere, where RIF has actually been used earlier? Or could the "upper mutation rate" of 2.2 SNPs/genome/year (Menardo et al 2019) be somewhat off?

Overall, I'm not convinced that the observation of a weird and very rare lineage which has retained a cobF gene region lost in other MTBC members, has any major consequences for interpretations of MTBC evolution and pathogenicity. Perhaps, with more data it will. However, I do believe it

further supports the notion of an African origin of MTBC, a notion which is already well established and uncontroversial. Additional specific comments follow:

1. The statement in the abstract and the paper that L8 is restricted to the great lakes region is too strong when there are only two observations available. The authors may well be correct that it is, but the evidence is still not available for such a strong statement.
2. The introduction starts with a statement that MTBC is among the most ancient human diseases, but I'm not convinced the sum of current evidence supports this notion. The TMRCA of MTBC has been estimated to 4-6000 years ago, using the best available (aDNA) methods (Bos et al Nature). Is this significantly older than "typical" pathogens?
3. Introduction line 11 first page: The authors fail to cite at least one important paper looking into the global spread of TB: O'Neill et al 2019 Mol Ecol. Rutaihwa et al 2019 Front Ecol Evol & Brynildsrud 2018 Sci Adv are additional studies which seem highly relevant in that they investigate the patterns and drivers of expansion of the two major lineages 2 and 4.
4. Second page of intro: what does a third main branch mean? Hard to decipher from phylogeny.
5. Second page of results: Please clarify what "screening global publicly available genome datasets" entails (methodology and tools used).
6. Figure 2 doesn't seem to add much value and could optionally be skipped entirely.

Reviewers' comments

Reviewer #1 (Remarks to the Author):

Jean Claude Semuto Ngabonziza and co-workers report the discovery of a novel, rare, geographically unique, human-to-human transmitted lineage (L-8) that is a phylogenetic outlier of canonical MTBC complex consisting of two major lineages. Complete genome-based investigation of a strain from this lineage explains the emergence, diversification, and success of MTB as a virulent and successful pathogen. The genomic resource and findings from this lineage fill a major and probably most critical gap in systematic understanding on the East African origin of MTBC from environmental or -non-professional pathogenic form as STB to an obligate, successful and virulent pathogen.

We thank the reviewer for the very positive comments on the significance of our results.

Below are major comments that need to be addressed before taking any decision on the manuscript

- 1) Why only *M. canettii* is used as an out-group? There is no statistical or phylogenetic support for the placement of L-8. Why can't it be part of second branch of MTBC (hence zoonotic origin) or in fact the common origin of *M. canettii* and L-8? The authors need to include additional species from STB (*M. kansasii*, *M. marinum*, etc) as outliers to support the intermediate position of L-8 between *M. canettii* and canonical MTBC.

Authors' reply: The phylogenetic support for the outgroup position of *M. canettii* is justified by combined lines of evidence, which we first indicate now as follows in lines 184-188: "*M. canettii* represents the closest outgroup to the MTBC including L8, as shown by subsequent comparative analysis of a complete L8 genome (see below), and previous observations of ~2.0 Mb larger genomes and substantially lower average nucleotide identities of

phylogenetically closest non-tuberculous mycobacterial species such as *M. marinum* and *M. kansasii*". We then indicate in lines 331-335: "In contrast, none of the almost 900 other genes specifically identified in the *M. canettii* genomes, and absent in the other MTBC genomes, were found in the L8 genome. The latter finding thus supports the close relationship with the previously known MTBC lineages indicated by the SNP-based phylogeny, as well as the outgroup position of *M. canettii* relatively to the MTBC including L8". We further clarify in lines 361-367: "The assembled L8 genome also included 48 out of 50 genes (the exceptions are rv3513c encoding the probable fatty-Acid-Coa ligase FadD18 and the PhiRv1 region) present in MTBC members but not found in any of the *M. canettii* genomes, including a number of genes putatively acquired through horizontal gene transfer by the common ancestor of the MTBC after its separation from *M. canettii* (Supplementary Table 5). This observation additionally supports both the close relationship with the previously known MTBC branches and the outgroup position of relatively to the MTBC including L8."

In addition to this, we have constructed a core gene alignment and subsequent phylogeny from representatives of the MTBC lineages, the Rwandan L8 genome, *M. canettii*, and *M. kansasii* and included this as supplementary figure 1. Due to the short branch lengths within the MTBC making visualisation of the topology difficult, we chose to show this as a cladogram where the branch lengths do not have meaning. This tree also resolved L8 as being separate from the rest of the MTBC and positioned between *M. canettii* and the other MTBC lineages, with 100% bootstrap support. The results and methods of this analyses have also been added to the main text (lines 191-194 and 750-756).

Between the two phylogenetic approaches and the gene gain/loss analysis, we believe that our finding of the placement of this new lineage to be robust.

2) How their study explains the origin of human and animal associated lineages...about host jump or about how human transmitted the disease to animals.

Authors' reply: Thank you for this question. In the Discussion, we explain that our results are consistent with the presumed scenario of a human rather than a zoonotic origin of the MTBC, already established 20 years ago in ref.31 and 43 as follows: “Moreover, the observation that both L8 strains share two uncommon rifampicin- and isoniazid-resistance conferring mutations in *rpoB* and *inhA* suggests that multidrug resistance was already acquired in the common ancestor of these two strains. Isoniazid and rifampicin were introduced in TB treatments in the African Great Lakes region in the late fifties and 1983, respectively (Dr. Armand Van Deun, personal communication). These shared MDR-defining mutations, and the detection of these isolates in human patients in both cases (with reported absence of previous TB history for the Rwandan patient), suggest that these patients were infected with an already-resistant strain, which was exposed to drug selective pressure already decades ago and had been likely circulating in the community for some time. Overall, this pattern thus suggests human-to-human transmission rather than infection from a non-human source. While based on only two initial strains, these results are consistent with the presumed scenario of a human rather than a zoonotic origin for the MTBC.” We respectfully think that speculating on host jump and how humans transmitted the disease to animals might be beyond the scope of this study.

3) The authors report that “Among 14 other isolates out of 27 from Uganda and Rwanda tested by Gene Xpert MTB/RIF that showed the same delayed probe B as L8, none displayed the L8 signatures when tested by Deeplex Myc-TB or by classical spoligotyping”. Since the initial phenotype was same as L-8 lineage strain and considering

outlier status of L-8, why authors have not investigated genome of these 14 or in fact 27 isolates or at least tested the presence or absence of *cobF* gene in these 27 isolates?

Authors' reply: Of the 27 samples tested by Gene Xpert that showed a delayed probe B, only 14 were available for further testing by Deeplex-MycTB or spoligotyping. This is clarified as follows: "Furthermore, among 14 other available isolates out of 27..." (line 221-223).

Delayed probe B "only" indicates the possible presence of a rifampicin resistance associated Asp435Tyr mutation in the *rpoB* gene, which can be generically found in rifampicin resistant strains of other lineages as well. To better suggest this, this section has been rephrased as follows: "The patient was diagnosed with rifampicin-resistant TB based on standard Xpert MTB/RIF testing (Xpert; Cepheid, Sunnyvale, CA, USA), which probes for mutations in the rifampicin resistance-determining region of the *rpoB* gene of MTBC strains. The results of the assay showed a rare delayed probe B reaction (~3% prevalence in Rwanda)¹⁵, presumed (and later confirmed; see below) to be due to the rifampicin resistance-associated Asp435Tyr mutation".

None of the available 14 isolates showed L8 SNP and/or spoligotype signatures, thus implying membership to other (known) MTBC lineages and not L8. As further explained in our responses to comments 5 and 6, beyond the analysis of 36 complete genomes representing previously known MTBC lineages, we further confirmed by BLAST analysis the systematic absence of *cobF* in any of 6,456 draft genome assemblies available as of January 2020 from the NCBI, from strains belonging to lineages 1 to 7 or the animal lineages of the MTBC. Given this systematic absence in thousands of other genomes from known MTBC lineages, combined with the clonality of the MTBC with negligible, if any, horizontal gene transfer between strains (see our responses to comments 5 and 6), it can be very

confidently inferred that *cobF* is almost undoubtedly also absent in the aforementioned 14 isolates.

4) What is the prevalence of *cobF* in sub-sample 1500 isolates originating from Uganda, Rwanda and DRC that were screened using Deeplex MTB or in the NCBI WGS data set of strains from MTBC complex?

Authors' reply: Likewise, no L8 SNP and spoligotype pattern was identified in these 1500 samples by Deeplex-MycTB, also implying membership to other known MTBC lineages. As mentioned above, none of the 36 complete genomes or the 6,456 quality draft genome assemblies from MTBC lineages 1 to 7 or animal lineages available from the NCBI was found to contain *cobF*. The only genome assembly in the NCBI dataset that we found to contain *cobF* is the L8 strain from a patient from Uganda that was originally misclassified as *M. bovis*, as indicated in the text.

5) In the context of their major finding, what is the status of *cobF* in the draft and the complete genomes of MTBC available in NCBI.

Authors' reply: *cobF* is lacking in all the complete genomes as well as in the draft genome assemblies other than the single L8 strain from Uganda available from the NCBI. Moreover, via BLAST analysis, we further confirmed the systematic absence of *cobF* in any of 6,456 quality draft genome assemblies available from the NCBI, from strains belonging to lineages 1 to 7 or the animal lineages of the MTBC. We thereby determined that the junction between the sequences flanking the *cobF* deletion was at the same nucleotide position in all but 6 of these genomes, resulting in the truncation of *rv0943c* and *rv0944* genes as seen in the complete MTBC genomes (Fig. 4). Consistent with the clonal evolution of the MTBC with negligible, if any, horizontal gene transfer between strains (ref. 1, 14, 32), this perfect conservation of this sequence junction suggests that *cobF* was lost in the MRCA of the other

MTBC lineages, after its divergence from L8. The 6 exceptions were 3 strains from lineage 4.3 and 3 that showed slightly larger deletions, including the 5' region of rv0943c or the 5' region of rv0943c, rv0944 and the 5' region of rv0945, respectively, suggesting probable subsequent deletion events in particular sub-branches of these lineages. This additional information is now included in Results as such (lines 327 onwards).

6) More importantly, what is the status of *pks8* and *cobF* gene region in Ugandan strain?

What is the colony morphology of Ugandan strain?

Authors' reply: We verified that the deletion of the *cobF* region and the *pks5* configuration were identical in the Ugandan strain. For *cobF*, this is clarified as follows: "Conversely, a particular 4.4 kb genome region was present in the genomes of both L8 strains and in *M. canettii* (line 300), and for *pks5* both L8 strains displayed the single polyketide-synthase-encoding *pks5* gene configuration shared by all MTBC members" (line 369). As indicated in lines 86-88, the Ugandan isolate was identified upon screening global, publicly available genome datasets. Despite contacts made with the involved Ugandan team, the corresponding strain was not available for phenotypic characterization, including colony morphology. This unavailability is now specified in Methods (line 560). However, given the identical *pks5* configuration shared with the Rwandan strain, we would have expected that this strain showed the same rough colony morphotype typical of other MTBC strains, also containing this single *pks5* gene configuration.

7) Genome quality statistics like completeness and coverage as estimated by softwares like CheckM is lacking.

Authors' reply: Thank you for this suggestion. A genome completeness check has been included using the CheckM lineage-specific workflow. This found a 98% completeness and

no contamination or strain heterogeneity. This has been outlined in the methods lines 739-742.

Moreover, we proceeded even more systematically by verifying the presence/absence of all genes or collocated genes (single- or multi-copy) in L8 versus 36 MTBC reference genomes descending from a same ancestor, as well in a phylogenetically proximal, external set of *M. canettii* genomes. As indicated in Methods, this was done by BLAST searches and analysis of gene synteny, using Artemis and Artemis Comparison Tool, combined with a custom Multiple Annotation of Genomes and Differential Analysis (MAGDA) software previously used for annotation of *M. canettii* and *Helicobacter pylori* genomes.

Apart from three deletions (RD3, RD5, RD14) that also arose independently in several other MTBC branches, and 24 repetitive/multicopy genes (IS6110-related, PE/PPE-, or Mce-encoding), we only found 5 non repetitive genes included in two small segments (3.4 and 4.4 kb), which were undetected in the complete L8 genome while being present in reference MTBC genomes (Supplementary Table 2). This is now indicated in Results as follows: “Apart from these three deletions and two dozen repetitive/multicopy genes (IS6110-related, PE/PPE-, or Mce-encoding), we only found 5 non-repetitive genes, included in two small segments (3.4 and 4.4 kb), which were undetected in the complete L8 genome while being present in reference MTBC genomes (Supplementary Table 2)”.

Conversely, the *cobF* locus is the only non-repetitive region that is present in the genomes of both L8 strains, and absent in all complete and draft MTBC genomes analysed (Supplementary Table 3). The two L8 genomes were sequenced completely independently, clearly excluding the result of a contamination, consistent with the CheckM results.

8) In the results section describing complete genome assembly of PacBio reads, coverage of the genome is not clearly mentioned. What does 186x, 39x and 38x infer? Are these corresponding to chromosome and plasmids in complete genome?

Authors' reply: These numbers refer to the estimated genome coverage depths using raw, corrected and trimmed PacBio reads, respectively. This is now clarified as follows in the Methods: "After discarding 60,272 reads below minimal quality parameters, 106,681 reads were used for the assembly. Based on the expected genome size, the average coverage depth was estimated at 186x using raw reads, and 39x and 38x using corrected and trimmed reads, respectively".

Other comments

- 1) In methods the last section it is mentioned as L9 genomes, it should be L8
- 2) Figure 1 legend L9 is mentioned.....it should be L8.

Authors' reply: Corrected as required

Reviewer #2 (Remarks to the Author):

This is a well-written paper reporting a very interesting and relevant finding for the TB community. In the context of two separate analyses (1 done in an MDR-TB trial, the other a screening of publicly available genomes), authors identified what appear to be a novel lineage of MTBC. The first strain, from Rwanda, was available to the authors for sequencing with PacBio/Illumina to complete the genome. The latter strain was from Uganda; the raw sample did not appear to be available for further analysis. The Rwandan and Ugandan strains appear very similar based on a variety of analyses (e.g., spoligo, comparison of AMR genes/mutations), and form a new clade on phylogenetic analysis compared to a representative set of global TB human and animal-adapted lineages.

Authors presented a comprehensive, detailed, and convincing analysis of these two strains, therefore, most of my comments are fairly minor.

Thank you for the appreciation of the interest and comprehensiveness of our analysis.

1) Last paragraph of the Intro – “we used PacBio and Illumina WGS to reconstruct the full circular genome” – please specify which sample (country) was used for this as well as why this was not done for both samples here. I understand only sequencing data was available for the second patient’s sample, but this was only clear much later in the paper; it would be more clear if this were stated upfront.

Authors’ reply: We accordingly changed this paragraph as follows: “These two strains were discovered in two independent analyses, and were both multidrug-resistant (MDR; i.e. resistant to at least rifampicin and isoniazid). One was isolated from a tuberculosis patient in Rwanda through an ongoing MDR-TB diagnostic trial in Africa. The second isolate was from a patient in Uganda, and its identification was inferred upon screening publicly available draft genome datasets, where it was misclassified as an *M. bovis* strain. We used

PacBio and Illumina WGS to reconstruct the full circular genome of the Rwandan strain. We utilized these data and the available Illumina sequencing data of the Ugandan strain to reconstitute the phylogeny of this novel lineage, which we named Lineage 8 (L8), and further investigate molecular and evolutionary events associated with the emergence of the MTBC.”

2) “Molecular scars”? Could authors please include a definition of this in the current manuscript, as even working in this field, this is not terminology I am familiar with; this definitely needs a bit more explanation to be accessible to a broader audience. Along these lines, will the general audience of this journal know what standard short-course MDR TB treatment is? Please review the manuscript for very TB-specific concepts, and whether these may warrant a bit of additional explanation for those who aren’t TB specialists.

Authors’ reply: Thank you for these suggestions. We now explained molecular scars as follows: “Further evidence for the early branching of L8 relative to the rest of the MTBC comes from examination of interrupted coding sequences (ICDSs). These ICDSs correspond to frameshifts or in-frame stop codons detected in genes originally intact in a common progenitor, thus putatively representing so-called molecular scars inherited during progressive pseudogenization of the MTBC genomes”. Likewise, we describe the standard short-course MDR TB treatment as follows: “standard short-course MDR-TB treatment (i.e. 9-month WHO-endorsed MDR-TB regimen, including moxifloxacin, kanamycin, protionamide, ethambutol, clofazimine, high dose isoniazid and pyrazinamide). Also in response to reviewer 1’s comments n°1, 3 and 5 (please see above), we further clarified other concepts or notions more specific to the TB field, such as the principle of the Xpert

diagnostic test, clonality of the pathogen's population structure with negligible horizontal gene transfer between strains, and the use of *M. canettii* as an outgroup.

3) There were a couple places where authors should be less firm about their conclusions, given the paucity of data (2 samples so far from this lineage); for example, in the Abstract, authors state Lineage 8 is restricted to the African Great Lakes region. They also state this on the second page after "WGS analysis" ("indicate that L8 is generally rare and geographically restricted to..."). While their two isolates are from this region, and the work is suggestive that this lineage is from this area, this is not conclusive from this work, so I would change the wording a bit here to convey this uncertainty. Another example is "our results indicate L8 is as clonal as the rest of the MTBC"; again, this is an analysis of 2 samples so this is very preliminary and suggestive at best.

Authors' reply: In accordance with your suggestions, we changed the abstract as follows: "...representing an as-yet-unknown lineage, named Lineage 8 (L8), seemingly restricted to the African Great Lakes region". Likewise, the part after WGS analysis was changed in: "The absence of any matching pattern in the global spoligotype database, as well as the lack of detection of this clade in previous large WGS datasets of MTBC strains from global sources, suggests that L8 is rare and seemingly geographically restricted to the African Great Lakes region". We also changed the last sentence as follows: "Although our analysis is limited to two genomes identified to date, our results suggest L8 is as clonal as the rest of the MTBC".

4) Related to this, please show the ClonalFrameML results in the Supplement, rather than saying 'data not shown'. Thank you.

Authors' reply: The graphical result of ClonalFrameML is now added in supplementary data (Supplementary Figure 2) and the exact number of recombination events (26) and the length of these events (maximum 607bp) is detailed in the results (lines 394-398).

5) Results – “L8 related TB patient in Rwanda” – is it ‘L8 related’ or is it just a ‘patient with L8 TB’? The wording could be changed here.

Authors’ reply: We changed this section title in: “Patient with the L8 TB strain in Rwanda”

6) Figure 2 - it says limits of detection are indicated in grey and orange at each position; this really wasn’t too obvious when looking at the figure – can authors please specify they mean the middle circle or otherwise indicate this in some way on the figure directly?

Authors’ reply: To clarify the localization of these grey and orange zones, the Figure 2 legend has been revised as follows: “Limits of detection (LOD) of potential heteroresistance (reflected by subpopulations of reads bearing a mutation), depending on the coverage depths over target sequence positions, are represented by grey (LOD 3%) and orange zones (variable LOD >3%–80%; only seen in extremities of a few targets, such as the two rrs regions) above the reference sequences within the sectors”.

7) There is three instances of "L9" being used instead of L8; please correct.

Authors’ reply: Corrected as required.

8) In the Discussion, authors report no previous TB treatment for the Rwandan patient. Have authors tried to contact the publishers of the Ugandan genome to see if this information would be available for this patient?

Authors’ reply: Indeed, we tried but unfortunately we could not obtain a response.

9) Also in the Discussion, “If true, more recently emerged or introduced cobF-deleted strains might have conceivably largely outcompeted L8 strains” – how would authors propose to test this hypothesis?

Authors’ reply: Of important note, very recent findings suggest an additional, plausible but not necessarily mutually exclusive hypothesis for this selective sweep, linked to another gene loss event in later branching MTBC lineages, involving the TbD1 region. We thus

explain it and proposed an experimental approach to test both hypotheses (i.e. related to *cobF* and TbD1) as follows: “As an additional plausible but not necessarily mutually exclusive hypothesis, such selective sweep of L8 might have been (further) enhanced by the loss of TbD1 region in later branching MTBC strains. This region, which we also found intact in L8, as is the case in the “ancestral” *M. tuberculosis* lineages L1 and L7, encodes members of the mycobacterial membrane protein families MmpL. Very recent findings indicate that the loss of this region in later branching MTBC strains was also associated with a gain of virulence, and the deletion of TbD1 at the origin of the “modern” *M. tuberculosis* lineages L2/L3/L4 has therefore been suggested as a key driver for their global epidemic spread. If true, more recently emerged or introduced *cobF*- and TbD1-deleted strains might conceivably have largely outcompeted L8 strains. This hypothesis could be explored by assessing the growth and the virulence/fitness in cellular and animal models, of recombinant *cobF*- and/or TbD1-knock-out, as well as *cobF*- and/or TbD1-knock-in strains, derived from the available Rwandan L8 strain and other MTBC strains, respectively” (line 528-532).

A few comments for clarity:

- 1) While I appreciate the format of the journal has the Methods last, many readers prefer to read the Methods before the Results to be able to properly interpret them. As the paper stands, some of the Methods are not clear on their own; for example, authors mention the Deeplex-MycTB assay in the Methods, but not what this is and what samples it is being done on, and don't explain that the Rwandan sample is actually avai. Could this please be added to the Methods themselves?

Authors' reply: In accordance with your suggestion, we now indicate in Methods lines 563-568: “Targeted sequencing was performed by using the Deeplex-MycTB assay (Genoscreen, France). Briefly, this assay relies on a 24-plexed PCR amplification of mycobacterial species

identification (hsp65), genotyping (spoligotyping and phylogenetic single nucleotide polymorphisms (SNPs)) and 18 *M. tuberculosis* complex drug resistance-associated gene targets. This test and short-read Illumina-based WGS were performed on the Rwandan strain as follows.”

2) In the Discussion “suggests prolonged exposure to antibiotic treatment and human-to-human transmission of a drug resistant strain” – This sentence could be rephrased for clarity. I think authors mean that these patients were likely infected with already-resistant strains, which had been likely circulating for some time in the community, not that these patients themselves had prolonged exposure to antibiotics? The following two sentences (starting with “The observation that...” could possibly be moved before this.

Authors’ reply: As suggested, we now indicate: “Moreover, the observation that both L8 strains share two uncommon rifampicin- and isoniazid-resistance conferring mutations in *rpoB* and *inhA* suggests that multidrug resistance was already acquired in the common ancestor of these two strains. Isoniazid and rifampicin were introduced in TB treatments in the African Great Lakes region in the late fifties and 1983, respectively (Dr. Armand Van Deun, personal communication). These shared MDR-defining mutations, and the detection of these isolates in human patients in both cases (with reported absence of previous TB history for the Rwandan patient), suggest that these patients were infected with an already-resistant strain, which was exposed to drug selective pressure already decades ago and had been likely circulating in the community for some time. Overall, this pattern thus suggests human-to-human transmission rather than infection from a non-human source”.

3) Please add line numbers and page to submissions if possible. It’s more difficult to keep track as a reviewer and properly refer the authors to the right sections without these. An extra space or indenting for new paragraphs is also helpful. Thank you.

Authors' reply: Done as suggested.

Reviewer #3 (Remarks to the Author):

The paper "A sister lineage of the Mycobacterium tuberculosis complex discovered in the African Great Lakes region" constitute a description of a new mycobacterium tuberculosis complex (MTBC) lineage which forms a taxonomic outgroup to the typically pathogenic lineages observed to date, but is more closely related to these and the possibly environmental *M. canettii*. The analyses performed strongly suggest that the new lineage (L8) is a very rare pathogen of humans. The observation is interesting, but in my opinion, of limited interest to people outside the core TB community, and would perhaps be more suitable for a more specialized journal.

Authors' comments: As of March, 25, our manuscript, which we deposited in Bio-archive on January, 20, has an Attention Score that is ranked in the top 5% of all research outputs scored by Altmetric (<https://biorxiv.altmetric.com/details/75079362>). Also accounting for the very positive appreciations of the two other reviewers, we believe that this provides objective evidence of the solid interest that our findings raise in and probably beyond the TB community.

The presence of resistance mutations yielding INH and RIF mutations which are both rare, yet common to both L8 isolates is quite interesting, especially in light of the significant SNP-distance between the two isolates (100 SNPs). Yet, it is also clear that the authors don't have much evidence at the moment to explain this observation which they state would indicate a high substitution rate (for the resistance SNPs to have emerged following the introduction of the relevant drugs in the region). In the discussion, the authors don't cite any source for their statement about the history of INH and RIF use in the region – could RIF

possibly have been used earlier? Could the isolates have emerged elsewhere, where RIF has actually been used earlier? Or could the “upper mutation rate” of 2.2 SNPs/genome/year (Menardo et al 2019) be somewhat off?

Authors’ comments: The date of introduction of RIF in TB treatments in the region has been updated to 1983, according to more precise information received from Dr. Van Deun of the Institute of Tropical Medicine in Antwerp, who was among the pioneers of treating TB with standardized regimen in the region (now cited as personal communication; see our response to reviewer 1’s comment 2). This therefore suggests a mutation rate in the range of this upper rate. In accordance, we indicate: “Of note, given the above timeline of introduction of rifampicin and isoniazid in both countries, the ~100 SNPs distance separating these two strains from their MRCA would imply a rapid molecular clock for L8, in the upper bound of 2.2 SNPs/genome/year most recently estimated for other MTBC clades”.

Overall, I’m not convinced that the observation of a weird and very rare lineage which has retained a *cobF* gene region lost in other MTBC members, has any major consequences for interpretations of MTBC evolution and pathogenicity. Perhaps, with more data it will. However, I do believe it further supports the notion of an African origin of MTBC, a notion which is already well established and uncontroversial.

Authors’ comments: As mentioned in Discussion, the discovery of this very rare lineage provides unique insights into an as yet unknown ancestral stage of the MTBC evolution and pathoadaptation. In particular, our findings reveal that the *pks5*-recombination-mediated surface remodelling, associated with a gain of virulence, preceded the loss of the *cobF* region, plausibly associated with a gain of fitness. Regarding the origin of the MTBC, our findings more specifically support this origin in the eastern part of the African continent, reinforcing results from previous work that were suggestive of this.

Additional specific comments follow:

1) The statement in the abstract and the paper that L8 is restricted to the great lakes region is too strong when there are only two observations available. The authors may well be correct that it is, but the evidence is still not available for such a strong statement.

Authors' reply: Done as suggested.

2) The introduction starts with a statement that MTBC is among the most ancient human diseases, but I'm not convinced the sum of current evidence supports this notion. The TMRCA of MTBC has been estimated to 4-6000 years ago, using the best available (aDNA) methods (Bos et al Nature). Is this significantly older than "typical" pathogens?

Authors' reply: Done as suggested.

3) Introduction line 11 first page: The authors fail to cite at least one important paper looking into the global spread of TB: O'Neill et al 2019 Mol Ecol. Rutaiwa et al 2019 Front Ecol Evol & Brynildsrud 2018 Sci Adv are additional studies which seem highly relevant in that they investigate the patterns and drivers of expansion of the two major lineages 2 and 4.

Authors' reply: Done as suggested.

4) Second page of intro: what does a third main branch mean? Hard to decipher from phylogeny.

Authors' reply: To simplify, this has been rephrased as follows: "Here, we describe two exceptional strains representing a new clade, diverging before the MRCA of the other MTBC lineages".

5) Second page of results: Please clarify what "screening global publicly available genome datasets" entails (methodology and tools used).

Authors' reply: Approximately 20,000 MTBC genomes, publicly available on NCBI/EBI were downloaded and processed through an in-house whole-genome sequencing pipeline developed for the analysis of MTBC genomes (described in the method section of our manuscript). From the analysis of these 20,000 MTBC genomes, we found 1 genome that did not classify in any of the 7 human-adapted lineages or 9 animal-adapted ecotypes known at the time. This is now indicated as follows: "The strain isolated in Uganda was discovered independently upon screening global MTBC genome datasets publicly available on NCBI/EBI, comprising approximately 20,000 genomes in total. From subsequent processing with our WGS analysis pipeline, we found 1 genome that did not classify in any of the 7 human-adapted lineages or 9 animal-adapted ecotypes known at the time, but had been misclassified as *M. bovis* isolated from a human patient" (line 160-172).

A phylogeny was then reconstructed to place the unclassified genome in the context of the known MTBC lineages, as described in the results.

6) Figure 2 doesn't seem to add much value and could optionally be skipped entirely.

Authors' reply: The Figure 2 was an output of Deeplex-MycTB assay, visualizing a very atypical genotype and spoligotype that was the base of Rwanda strain discovery, thus very relevant to this study. As such, we think important to keep it as a main figure.

REVIEWERS' COMMENTS:

Reviewer #1 (Remarks to the Author):

All the comments are adequately addressed in the revised version.